# Effect of supplemental nutrition in pregnancy on offspring's risk of cardiovascular disease in young adulthood: Long-term follow-up of a cluster trial from India

Sanjay Kinra[1]*, John Gregson[1], Poornima Prabhakaran[2], Vipin Gupta[3], Gagandeep Kaur Walia[2], Santhi Bhogadi[4], Ruby Gupta[2], Aastha Aggarwal[2], Poppy Alice Carson Mallinson[1], Bharati Kulkarni[4], Dorairaj Prabhakaran[2], George Davey Smith[5], K. V. Radha Krishna[4], Shah Ebrahim[1], Hannah Kuper[1], Yoav Ben-Shlomo[6]

1 Department of Non-Communicable Disease Epidemiology, London School of Hygiene & Tropical Medicine, London, United Kingdom, 2 Public Health Foundation of India, Delhi, India, 3 Department of Anthropology, University of Delhi, Delhi, India, 4 National Institute of Nutrition, Hyderabad, India, 5 MRC Integrative Epidemiology Unit, University of Bristol, Bristol, United Kingdom, 6 Population Health Sciences, University of Bristol, Bristol, United Kingdom

* Sanjay.Kinra@lshtm.ac.uk

**Data Availability Statement:** Data cannot be shared publicly because of ethical restrictions. Data are available from the APCAPS cohort (contact via

## Abstract

### Background

Undernutrition during intrauterine life and early childhood is hypothesised to increase the risk of cardiovascular disease (Developmental Origins of Health and Disease Hypothesis), but experimental evidence from humans is limited. This hypothesis has major implications for control of the cardiovascular disease epidemic in South Asia (home to a quarter of world's population), where a quarter of newborns have low birth weight. We investigated whether, in an area with prevalent undernutrition, supplemental nutrition offered to pregnant women and their offspring below the age of 6 years was associated with a lower risk of cardiovascular disease in the offspring when they were young adults.

### Methods and findings

The Hyderabad Nutrition Trial was a community-based nonrandomised controlled intervention trial conducted in 29 villages near Hyderabad, India (1987–1990). Protein-calorie food supplement was offered daily to pregnant and lactating women (2.09 MJ energy and 20–25 g protein) and their offspring (1.25 MJ energy and 8–10 g protein) until the age of six years in the 15 intervention villages, but not in the 14 control villages. A total of 1,826 participants (949 from the intervention villages and 877 from the control villages, representing 70% of the cohort) at a mean age of 21.6 years (62% males) were examined between 2009 and 2012. The mean body mass index (BMI) of the participants was 20 kg/m$^2$ and the mean systolic blood pressure was 115 mm Hg. The age, sex, socioeconomic position, and urbanisation-adjusted effects of intervention (beta coefficients and 95% confidence intervals)

http://apcaps.lshtm.ac.uk/) for researchers who meet the criteria for access to confidential data.

**Funding:** This research was funded by the Wellcome Trust, UK (https://wellcome.ac.uk/), through grant numbers 083707 (HK, SK, KVRK SE, YB-S), 084774 (SE, GDS, SK), and 084754 (SE, SK, PP). GDS works in the Medical Research Council Integrative Epidemiology Unit at the University of Bristol MC_UU_00011/1. The funders had no role in study design, data collection and analysis, decision to publish, or preparation of the manuscript.

**Competing interests:** I have read the journal's policy and the authors of this manuscript have the following competing interests: GDS is a member of the Editorial Board of *PLOS Medicine*.

**Abbreviations:** APCAPs, Andhra Pradesh Children and Parents study; BMI, body mass index; HOMA-IR, homeostatic model assessment-insulin resistance; ICDS, Integrated Child Development Services; LDL, low-density lipoprotein; LMIC, low- and middle-income country; LSHTM, London School of Hygiene & Tropical Medicine; MET, metabolic equivalent task; NTLI, nighttime light intensity; SLI, Standard of Living Index.

on outcomes were as follows: carotid intima-media thickness, 0.01 mm (−0.01 to 0.03), $p = 0.36$; arterial stiffness (augmentation index), −1.1% (−2.5 to 0.3), $p = 0.097$; systolic blood pressure, 0.5 mm Hg (−0.6 to 1.6), $p = 0.36$; BMI, −0.13 kg/m$^2$ (−0.75 to 0.09), $p = 0.093$; low-density lipoprotein (LDL) cholesterol, 0.06 mmol/L (−0.07 to 0.2), $p = 0.37$; and fasting insulin (log), −0.06 mU/L (−0.19 to 0.07), $p = 0.43$. The limitations of this study include nonrandomised allocation of intervention and lack of data on compliance, and potential for selection bias due to incomplete follow-up.

## Conclusions

Our results showed that in an area with prevalent undernutrition, protein-calorie food supplements offered to pregnant women and their offspring below the age of 6 years were not associated with lower levels of cardiovascular risk factors among offspring when they were young adults. Our findings, coupled with evidence from other intervention studies to date, suggest that policy makers should attach limited value to cardiovascular health benefits of maternal and child protein-calorie food supplementation programmes.

## Author summary

### Why was the study done?

- Some researchers say that poor nutrition of the mother in pregnancy can increase the risk of heart disease in her children.

- If true, this theory could help to explain the high risk of heart disease among South Asians, in whom low birth weight is very common.

- Evidence for this theory from intervention studies (in which some people are exposed to treatment and others are not) is limited.

### What did the researchers do and find?

- The Hyderabad Nutrition Trial was conducted previously (1987–1990) in an area with high levels of undernutrition in Telangana state, India; extra food (500 calories and 20 g protein) was offered daily to all pregnant women and their children below the age of 6 years in 15 villages out of a total of 29 villages.

- The researchers examined 1,826 children born during this trial when their average age was 22 years.

- They found no difference in risk of heart disease between offspring born to women who were offered extra food in pregnancy and childhood and those who were not.

### What do these findings mean?

- Our results showed that extra food offered to undernourished pregnant women may not lower their offspring's risk of heart disease in young adulthood.

- Our findings suggest that, until further evidence is available, policy makers should attach limited value to heart health benefits of food supplementation programmes for pregnant women and young children.

## Introduction

Cardiovascular disease is the leading cause of death and disability worldwide. The striking rise in the prevalence of cardiovascular disease in many low- and middle-income countries (LMICs), notably South Asia, is unexplained by population ageing and lifestyle changes associated with economic transition [1,2]. Social disadvantage in the early years of life appears to account for some of the increase in risk, but the biological mechanisms for this association are unknown [3]. A series of reports linking low birth weight to cardiovascular disease, as well as supportive animal models, have led to the hypothesis that undernutrition during intrauterine life and early childhood (critical or sensitive period) may have long-lasting effects on the future risk of cardiovascular disease through persistence of metabolic and physiological adaptations (Developmental Origins of Health and Disease Hypothesis) [4–7]. However, most of these studies were conducted in high-income countries, where maternal smoking during pregnancy is a more common cause of low birth weight than undernutrition, leading other researchers to suggest that the observed associations may be confounded by behavioural risks associated with persistent social disadvantage [8,9].

Few studies have been able to test this hypothesis in a trial setting, and the findings have been inconsistent [10–12]. We assessed cardiovascular disease risk factors among a cohort of young adults who were born during a trial of protein-calorie supplemental nutrition offered to pregnant women and young children in an area with prevalent undernutrition. We hypothesised that supplemental nutrition during pregnancy and early childhood would be associated with lower levels of cardiovascular disease risk factors in young adulthood.

## Methods

The study is reported in accordance with the CONSORT guideline (S1 CONSORT checklist). The Hyderabad Nutrition Trial (1987–1990) was a community-based nonrandomised controlled intervention trial conducted in 29 villages of Ranga Reddy district in Telangana state, India [12,13]. It evaluated India's Integrated Child Development Services (ICDS) scheme, which is a long-standing scheme aimed at improving child growth and development through integrated provision of food supplementation, anaemia control, immunisation, education, and basic healthcare to pregnant and lactating women and children up to 6 years [13]. Using the opportunity afforded by the gradual rollout of this nationwide scheme during the 1980s and 1990s, the National Institute of Nutrition of India conducted the trial to assess the effect of food supplementation in pregnancy on the offspring's birth weight. A cluster of villages was chosen from two adjacent administrative areas (called 'blocks'): one with the ICDS scheme in place (intervention arm) and the other awaiting implementation at that time (control arm) (Fig 1). As the 100 or so villages in each of the two blocks were spread over an unfeasibly large area for data collection, contiguous villages surrounding the geographic centre of each block were selected to make up the planned sample size of 30,000 total population in each block. This resulted in 15 intervention and 14 control villages geographically separated by uninvolved villages. The food supplement ('upma', a local food prepared from corn-soya blend and soya bean oil) was offered daily to women throughout pregnancy and lactation (2.51 MJ of energy

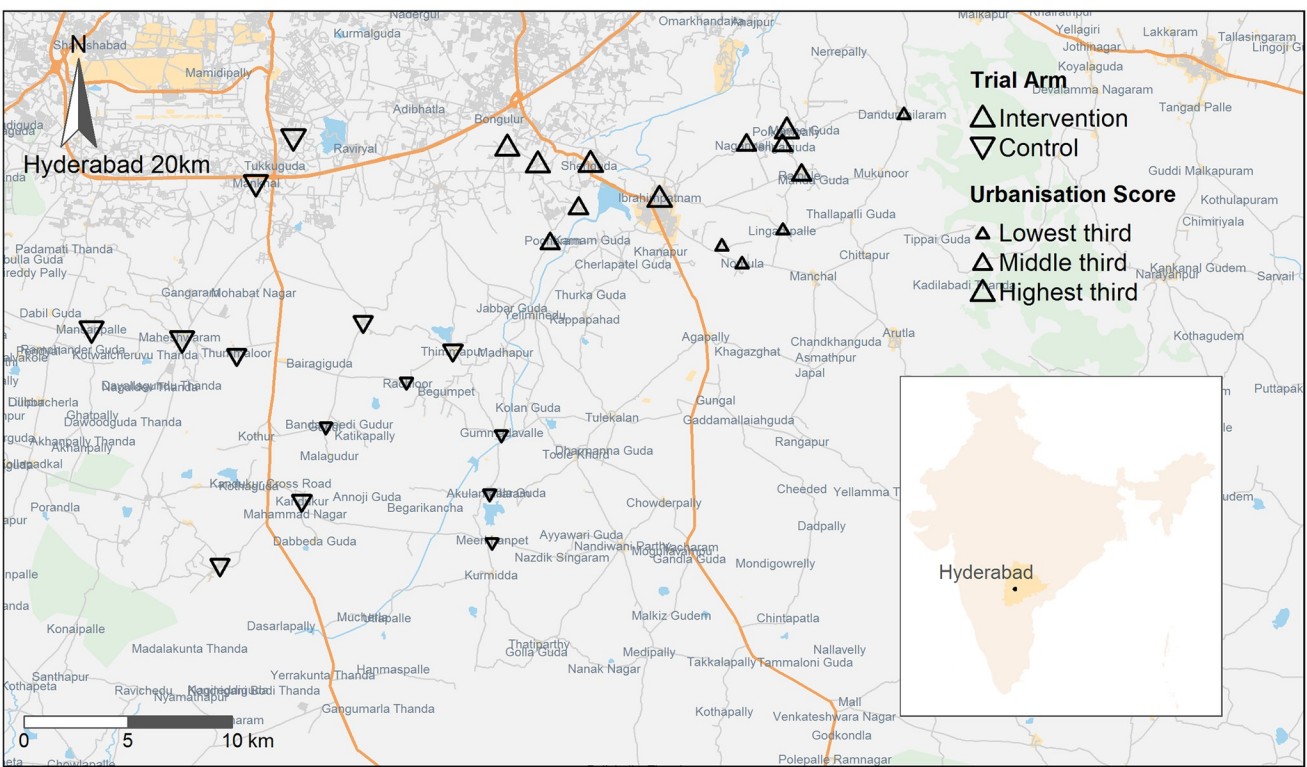

**Fig 1. Map of APCAPs villages.** Village urbanisation measured by nighttime light intensity assessed by remote sensing. Background maps were rendered by the authors using R, based on geodata made available by Geofabrik.de and OpenStreetMap.org, under the Open Database License (https://opendatacommons.org/licenses/odbl/1.0/). APCAPs, Andhra Pradesh Children and Parents study.

and 20–25 g of protein daily) and children below the age of 6 years (1.25 MJ and 8–10 g protein daily). A total of 2,964 birth weights were recorded within 48 hours of delivery with an infant beam balance with an accuracy of 20 g (John Chatillon & Sons, NY). The mean birth weight of children born in the intervention area was higher than control (2,655 g versus 2,594 g), with a mean difference of 61 g (95% confidence interval 18 to 104; $p$ = 0.007) [12,14]. The trial was not registered, as it was not standard practice to register community trials at that time. The trial families were recontacted in 2003–2005; offspring born in the trial ($N$ = 2,601) and their parents and siblings constitute the intergenerational Andhra Pradesh Children and Parents study (APCAPs) [15].

The offspring born during the trial have been followed up three times: first follow-up in 2003–2005 (mean age 16 years; $N$ = 1,165), second follow-up in 2009–2010 (mean age of 20 years; $N$ = 1,446), and third follow-up in 2010–2012 (mean age of 22 years; $N$ = 1,360) (Fig 2). Fewer participants were invited for examination at the first follow-up ($N$ = 1,492, 78% response) due to resource constraints (it was a PhD project funded by a small travel award); priority was given to those on whom additional data on compliance and growth were thought to be available at that time. At the time of the first follow-up at age 16 years, children born in the intervention villages were found to be taller and had lower levels of arterial stiffness and insulin resistance, suggesting lower risk of cardiovascular disease [12]. In this report, we present the findings from the second and third follow-ups of trial offspring. The third follow-up was conducted primarily to extend the study to parents and siblings of the trial offspring; however, the trial offspring were also offered the opportunity to take part again, resulting in the close timing of the two surveys. Given the close timing and a substantial number of

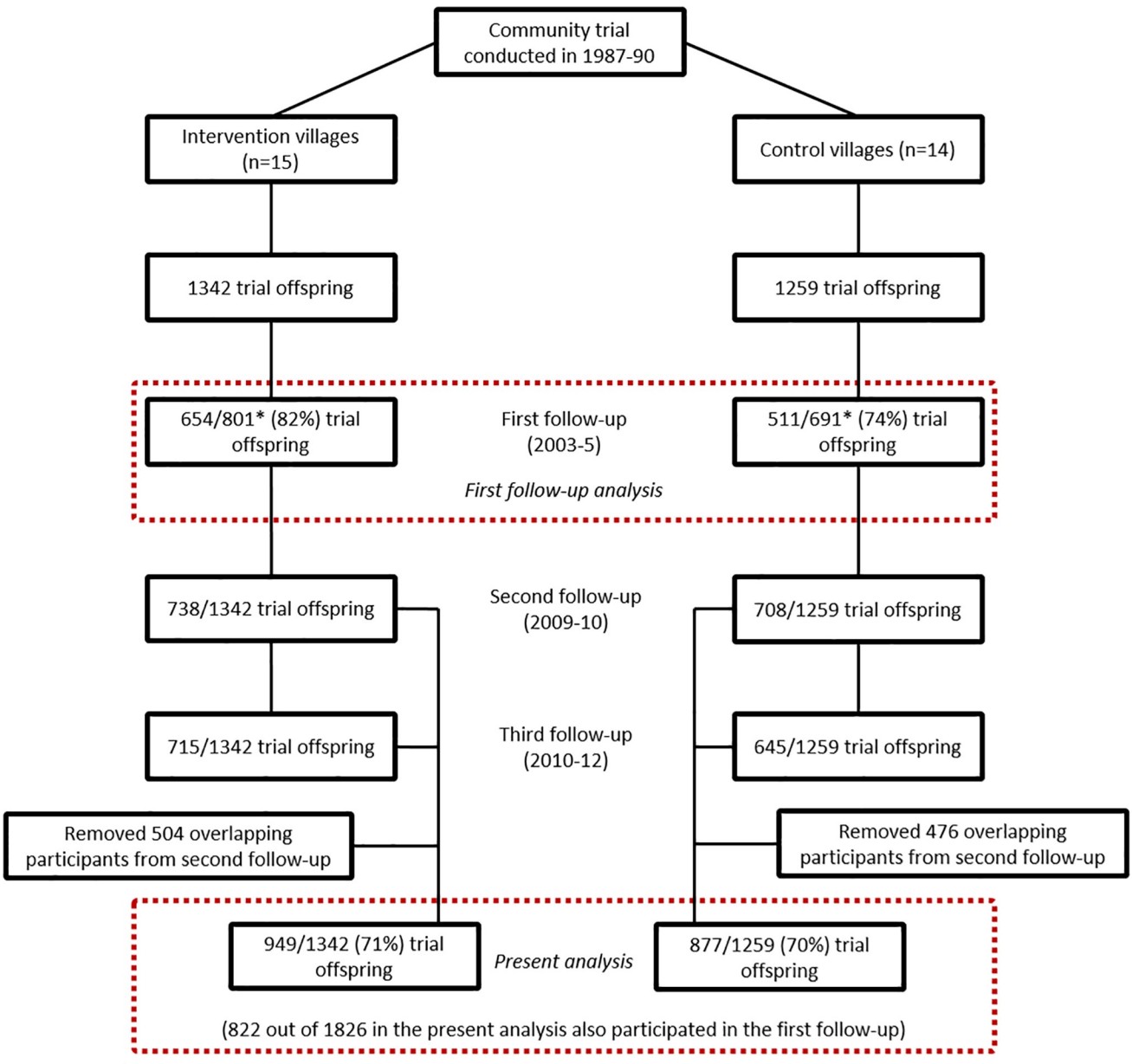

**Fig 2. Flowchart of APCAPs study participants at the first, second, and third follow-ups, 2003–2012.** APCAPs, Andhra Pradesh Children and Parents study.

nonoverlapping participants, we combined data from the two follow-ups to maximise study power and reduce selection bias. Where individual participant data were available from both surveys, we used data from the later survey wave because our primary interest was in cardiovascular risk in adulthood. This resulted in 1,826 participants at a mean age of 21.6 years (62% males). Since the first follow-up, the peri-urban study villages have experienced rapid unplanned urbanisation, which we quantified using data from remote sensing and took into account in our analyses (see details later) [16]. Ethical approvals for second and third follow-ups of the cohort were obtained from the Public Health Foundation of India, New Delhi,

India; National Institute of Nutrition, Hyderabad, India; University of Bristol, Bristol, United Kingdom; and the London School of Hygiene & Tropical Medicine (LSHTM), London, UK. Verbal permissions were taken from heads and governing committees of the villages. Written informed consent (or witnessed thumbprint if illiterate) for inclusion in the study was obtained from each participant prior to enrolment.

## Measurements

The study participants were interviewed and examined by trained observers using standardised methods, which have been described in detail previously (full protocol in S1 Text) [15]. No change was made to the study protocol after the start of the study. Sociodemographic and lifestyle (i.e., tobacco and alcohol use) data were collected using standard questions from India's Third National Family Health Survey [16]. Socioeconomic position was assessed using a subset of 14 questions (out of 29) of the Standard of Living Index (SLI) and applying the prescribed weights; a higher score indicates a higher socioeconomic position. A semiquantitative food frequency questionnaire coupled with customised nutrient databases was used to estimate average daily salt and fat consumption and energy intake; its validation against multiple weighed 24-hour recalls in this setting has been published [17]. Physical activity was assessed using a validated questionnaire (activities over the past week), which was used to derive metabolic equivalent tasks (METs; expressed in hours/day) and time spent sedentary (minutes/day); its validation against triaxial accelerometers in this setting has been published [18]. Anthropometric assessments were carried out using standard protocols [15]. Weight was measured with a digital weighing scale (SECA, www.seca.com) and standing height with a plastic stadiometer (Leicester measure; Chasmors, London, UK). Waist circumference was measured using a non-stretch metallic tape at the narrowest point of the abdomen between the ribs and the iliac crest. Anthropometric measurements were taken twice and averaged for analyses; where the difference between readings was more than the acceptable level (5 mm for height, 0.5 kg for weight, and 1 cm for waist circumference), a third reading was taken.

Vascular assessments were carried out using recommended guidelines, as described previously [19]. Blood pressure was measured on the right arm in the supine position using a validated oscillometric device (Model M5-I; Omron, Matsusaka, Japan) with appropriate cuff size. Arterial stiffness (augmentation index and aortic pulse wave velocity) and central systolic blood pressure were assessed using a Vicorder device (Skidmore Medical, Bristol, UK); higher values for arterial stiffness measures indicate stiffer arteries and increased vascular age. Blood pressure cuffs were placed around the neck (at the level of carotid artery) and upper thigh and inflated to 60 mm Hg before recording the waveforms. Three consecutive measurements of blood pressure and arterial stiffness were taken after a rest of five minutes (one minute of rest between readings) and then averaged for analyses. Carotid intima media thickness was measured on the near wall of the distal common carotid artery (right-hand side) using a B-mode ultrasound (Ethiroli Tiny 16a; Surabhi Biomedical Instrumentation, Chennai, India). Images were normalised before analyses by Carotid Plaque Texture Analysis software (LifeQ, Engomi, Cyprus). Two measurements of carotid intima media thickness were taken by separate operators and averaged for analysis.

Venous blood samples were collected after an overnight fast. Glucose was assayed on the same day using the oxidase-peroxidase enzymatic (GOD-PAP) method (reagents supplied by Randox Laboratory, Crumlin city, UK). An enzymatic colorimetric method was used to assay total cholesterol, triglycerides, and serum high-density lipoprotein cholesterol (HDL-C). Lipids and insulin assays were carried out using reagents supplied by Roche Diagnostics, Basel, Switzerland. Serum insulin was assayed on an e-411 autoanalyser using an electrochemiluminescence

immunoassay. The quality of biochemical assays was assured through internal controls and external assurance arrangements with Randox International Quality Assessment Scheme (lipids) and UK National External Quality Control Assessment Service (insulin). Intra- and inter-assay coefficients of variation were ≤3% and ≤5%, respectively, for all assays.

Quality of clinical measurements was ensured through rigorous protocols, regular standardisation of equipment, and teams to detect any drifts over time [15]. Reproducibility of measurements was evaluated by repeat measurements on a 5% random subsample; the intraclass correlation coefficients were >0.98 for anthropometric measurements, >0.85 for vascular measurements, and >0.94 for biochemical assays.

Village urbanisation was quantified using nighttime light intensity (NTLI), a data product derived from satellite sensors that capture visible near-infrared emissions from the earth's surface, which has been used to measure urbanisation and predict cardio-metabolic risk in several settings, including APCAPs cohort participants [19,20]. Geocoded village boundaries were applied to 2012 data from the Defense Meteorological Satellite Program (Operational Linescan System) accurate to a 1-km resolution, such that NTLI values represent emissions from the village areas only.

### Statistical analyses

We compared values of cardiovascular risk factors in intervention and control villages using a linear regression model and assessed the effect of the intervention using beta coefficients. We fitted a random intercept at both village and household levels, to allow for clustering of outcomes. Adiposity was estimated by body mass index (BMI); pulse pressure as difference between systolic and diastolic blood pressure; low-density lipoprotein (LDL) cholesterol using the Friedewald-Fredrickson formula; and insulin resistance by homeostatic model assessment-insulin resistance (HOMA-IR). We log-transformed the following variables prior to analyses to achieve a more Gaussian distribution: triglycerides, insulin, and HOMA-IR. Values of insulin were standardised across the two waves in order to allow for differences in assay techniques. Analyses were carried out using the 'intention-to-treat' principle, comparing risk factors of children born in the intervention and control areas. In view of the nonrandomised trial design and varying urbanisation of the study villages over the long-term follow-up, we fitted three predefined models to incrementally adjust for potential differences in sociodemographic and lifestyle factors that could influence the levels of cardiovascular risk factors: model 1 was adjusted for age and sex; model 2 was additionally adjusted for socioeconomic position and village urbanisation; and model 3 was further adjusted for lifestyle factors, including tobacco and alcohol use, fat and salt consumption, and physical activity level and sedentariness. We used formal tests of interactions to examine whether the effects of membership of treatment arm on outcomes were modified by age (split by median age of 22 years), sex, village urbanisation (thirds) and current obesity (split by median of 20 kg/m$^2$). To assess the impact of potential difference in samples between first and second or third follow-ups, we restricted the analyses to those who also presented at the first follow-up (sensitivity analyses).

### Results

A total of 1,826 (70% of cohort) participants were examined at least once between 2009 and 2012, of which 881 were also examined during the first follow-up in 2003–2005. In comparisons of cardiovascular risk factor data collected in the first follow-up, those who took part in the second or third follow-ups were broadly similar to those who took part in the first follow-up alone (Table 1).

**Table 1. Participant characteristics in 2003–2005 split by presentation in 2009–2012.**

| Characteristics | Absent in 2009–2012 (*n* = 284) | Present in 2009–2012 (*n* = 881) | *p*-Value |
|---|---|---|---|
| **ALL PARTICIPANTS** | | | |
| From intervention village | 154 (58.6%) | 475 (57.9%) | 0.84 |
| Age (years) | 15 (1) | 15 (1) | 0.94 |
| Male sex | 60 (22.8%) | 529 (64.4%) | <0.001 |
| Village urbanisation score | | | 0.96 |
| Least urbanised third | 71 (27.1%) | 233 (28.4%) | |
| Middle third | 91 (34.7%) | 277 (33.8%) | |
| Most urbanised third | 100 (38.2%) | 310 (37.8%) | |
| SLI score | | | <0.001 |
| Bottom third | 57 (21.5%) | 300 (36.6%) | |
| Middle third | 90 (34.0%) | 263 (31.3%) | |
| Top third | 118 (44.5%) | 257 (31.3%) | |
| Occupation | | | <0.001 |
| Unemployed | 25 (9.2%) | 60 (7.2%) | |
| Student | 203 (74.4%) | 703 (83.9%) | |
| Employed | 45 (16.5%) | 75 (8.9%) | |
| Height (cm) | 149.1 (14.1) | 150.1 (7.6) | 0.18 |
| BMI (kg/m$^2$) | 20.7 (4.4) | 20.7 (4.2) | 0.85 |
| Systolic blood pressure (mm Hg) | 108.3 (9.2) | 109.2 (10.5) | 0.20 |
| Total cholesterol (mmol/L) | 3.6 (0.7) | 3.4 (0.7) | <0.001 |
| Insulin (mU/L) | 19.0 (12.1 to 26.4) | 16.8 (10.9 to 24.9) | 0.01 |
| Augmentation index (%) | 5.0 (−2.5 to 10.5) | 5.0 (−3.5 to 11.0) | 0.67 |
| **CONTROL PARTICIPANTS** | | | |
| From intervention village | 0 (0.0%) | 0 (0.0%) | |
| Age (years) | 15 (1) | 15 (1) | 0.28 |
| Male sex | 28 (25.7%) | 225 (65.0%) | <0.001 |
| Village urbanisation score | | | 0.001 |
| Least urbanised third | 39 (35.8%) | 141 (40.8%) | |
| Middle third | 45 (41.3%) | 82 (23.7%) | |
| Most urbanised third | 25 (22.9%) | 123 (35.5%) | |
| SLI score | | | <0.001 |
| Bottom third | 19 (18.4%) | 127 (39.9%) | |
| Middle third | 39 (37.9%) | 102 (32.1%) | |
| Top third | 45 (43.7%) | 89 (28.0%) | |
| Occupation | | | 0.06 |
| Unemployed | 10 (9.7%) | 34 (10.6%) | |
| Student | 75 (72.8%) | 258 (80.4%) | |
| Employed | 18 (17.5%) | 29 (9.0%) | |
| Height (cm) | 149.7 (6.2) | 150.2 (6.0) | 0.56 |
| Body mass index (kg/m$^2$) | 20.8 (5.0) | 20.8 (4.1) | 1.00 |
| Systolic blood pressure (mm Hg) | 109.6 (8.7) | 109.3 (10.2) | 0.80 |
| Total cholesterol (mmol/L) | 3.5 (0.6) | 3.4 (0.7) | 0.036 |
| Insulin (mU/L) | 21.3 (14.1 to 30.8) | 18.1 (12.5 to 26.9) | 0.038 |
| Augmentation index (%) | 6.3 (2.5 to 10.3) | 6.5 (−1.5 to 12.0) | 0.73 |
| **INTERVENTION PARTICIPANTS** | | | |
| From intervention village | 154 (100.0%) | 475 (100.0%) | |
| Age (years) | 15 (1) | 15 (1) | 0.41 |

*(Continued)*

**Table 1.** (Continued)

| Characteristics | Absent in 2009–2012 (*n* = 284) | Present in 2009–2012 (*n* = 881) | *p*-Value |
|---|---|---|---|
| Male sex | 32 (20.8%) | 304 (64.0%) | <0.001 |
| Level of urbanisation (thirds) | | | 0.042 |
| Least urbanised | 32 (20.9%) | 92 (19.4%) | |
| Middle third | 46 (30.1%) | 195 (41.1%) | |
| Most urbanised | 75 (49.0%) | 187 (39.5%) | |
| SLI score | | | 0.002 |
| Bottom third | 30 (21.0%) | 156 (32.5%) | |
| Middle third | 45 (31.5%) | 137 (30.9%) | |
| Top third | 68 (47.6%) | 150 (33.9%) | |
| Occupation | | | 0.036 |
| Unemployed | 12 (8.0%) | 22 (4.8%) | |
| Student | 116 (77.3%) | 394 (86.2%) | |
| Employed | 22 (14.7%) | 41 (9.0%) | |
| Height (cm) | 148.7 (17.2) | 150.0 (8.6) | 0.24 |
| Body mass index (kg/m$^2$) | 20.7 (4.1) | 20.6 (4.3) | 0.79 |
| Systolic blood pressure (mm Hg) | 107.4 (9.4) | 109.2 (10.7) | 0.067 |
| Total cholesterol (mmol/L) | 3.7 (0.7) | 3.4 (0.7) | <0.001 |
| Insulin (mU/L) | 16.7 (11.3 to 25.2) | 15.6 (10.2 to 22.3) | 0.090 |
| Augmentation index (%) | 2.5 (−5.3 to 11.3) | 3.0 (−5.0 to 10.5) | 0.82 |

Data shown are number (%), mean (standard deviation), or median (interquartile range).

Data shown are only for those who took part in the 2003–2005 follow-up.

Abbreviations: BMI, body mass index; SLI, Standard of Living Index

The mean age of the participants was 21.6 years (62% males) (Table 2). Just over half of the participants were employed and about a quarter were in full-time education. About 10% consumed tobacco and mean BMI was 20 (SD 3.2) kg/m$^2$. The village urbanisation score was clearly different between trial arms: about 20% of the intervention arm participants lived in the least urbanised third of villages, as opposed to 44% in the control arm (3 villages in intervention arm and 7 villages in control arm belonged to the least urbanised thirds category). Missing data, notably for arterial stiffness and carotid intima media thickness measurements (relatively more due to greater time commitment required from the participants), were evenly distributed across the trial arms. There were no differences in the levels of cardiovascular risk factors between trial arms (Table 3). There was no strong evidence for effect modification by age, sex, village urbanisation and current obesity (see S1, S2, S3 and S4 Tables). In sensitivity analyses restricted to participants who were also examined in the first follow-up, results were broadly similar; although borderline effects of intervention on some outcomes (height, augmentation index, and HOMA score) were noted in models adjusted for socioeconomic position and urbanisation, their confidence intervals overlapped with those for effect estimates for the full cohort (Table 4).

## Discussion

In this long-term follow-up of a community-based nonrandomised controlled intervention trial conducted in an area with prevalent undernutrition, provision of supplemental nutrition to pregnant and lactating women (2.09 MJ energy and 20–25 g protein daily) and their

**Table 2. Distribution of cardiovascular risk factors of APCAPs participants in 2009–2012.**

| Risk factors | Missing data* (%) | Intervention (*n* = 949) | Control (*n* = 877) | *p*-Value |
|---|---|---|---|---|
| Age (years) | 0.0% | 21.7 (1.8) | 21.6 (1.9) | 0.79 |
| Male | 0.0% | 580 (61.1%) | 530 (60.4%) | 0.78 |
| Village urbanisation score | 0.4% | | | 0.32 |
| Least urbanised third | | 185 (19.7%) | 381 (43.4%) | |
| Middle third | | 400 (42.5%) | 202 (23.0%) | |
| Most urbanised third | | 356 (37.8%) | 294 (33.5%) | |
| Occupation | 0.0% | | | 0.26 |
| Unemployed | | 209 (22.0%) | 188 (21.4%) | |
| Student | | 259 (27.3%) | 202 (23.0%) | |
| Employed | | 481 (50.7%) | 487 (55.5%) | |
| SLI score | 0.5% | | | 0.65 |
| Bottom third | | 208 (22.0%) | 215 (24.6%) | |
| Middle third | | 336 (35.6%) | 300 (34.4%) | |
| Top third | | 400 (42.4%) | 358 (41.0%) | |
| Tobacco use (current/former) | 0.1% | 117 (12.3%) | 90 (10.3%) | 0.18 |
| Alcohol use (daily) | 0.1% | 25 (2.6%) | 22 (2.5%) | 0.82 |
| Time spent sedentary (hours/day) | 1.4% | 5.5 (3.6 to 7.9) | 5.6 (3.5 to 8.5) | 0.95 |
| Physical activity (MET-hours/day) | 1.5% | 36.4 (34.1 to 39.0) | 36.3 (34.0 to 39.6) | 0.44 |
| Salt consumption (g/day) | 0.2% | 2.0 (1.7 to 2.3) | 2.0 (1.7 to 2.2) | 0.34 |
| Fat consumption (% energy intake) | 0.2% | 18.8 (5.4) | 18.6 (5.5) | 0.51 |
| Height (mm) | 0.0% | 1616.6 (90.3) | 1614.8 (90.4) | 0.73 |
| BMI (kg/m$^2$) | 0.0% | 20.1 (3.3) | 20.1 (3.1) | 0.44 |
| Waist circumference (mm) | 0.2% | 691.7 (86.6) | 693.5 (81.6) | 0.43 |
| Systolic blood pressure (mm Hg) | 0.1% | 115.2 (10.4) | 114.2 (10.0) | 0.23 |
| Diastolic blood pressure (mm Hg) | 0.1% | 73.5 (9.7) | 72.2 (9.5) | 0.088 |
| Pulse pressure (mm Hg) | 0.1% | 41.6 (6.2) | 42.0 (6.3) | 0.45 |
| Augmentation index (%) | 26.1% | 12.9 (7.1) | 14.5 (7.5) | 0.086 |
| Central systolic BP (mm Hg) | 22.0% | 106.4 (9.4) | 106.9 (9.6) | 0.48 |
| Pulse wave velocity (m/sec) | 13.7% | 6.0 (0.7) | 5.9 (0.6) | 0.23 |
| Carotid IMT (mm) | 33.1% | 0.55 (0.11) | 0.56 (0.12) | 0.75 |
| Total cholesterol (mmol/L) | 1.0% | 4.1 (1.0) | 4.0 (0.8) | 0.28 |
| LDL cholesterol (mmol/L) | 1.4% | 2.4 (0.8) | 2.3 (0.7) | 0.12 |
| HDL cholesterol (mmol/L) | 1.0% | 1.1 (0.3) | 1.1 (0.3) | 0.56 |
| Triglycerides (mmol/L) | 1.0% | 1.1 (0.8 to 1.4) | 1.0 (0.8 to 1.4) | 0.41 |
| Fasting glucose (mmol/L) | 1.0% | 4.9 (4.5 to 5.2) | 4.8 (4.5 to 5.1) | 0.94 |
| Insulin (mU/L) | 1.4% | 6.2 (4.0 to 9.5) | 5.8 (3.6 to 8.9) | 0.49 |
| HOMA-IR | 1.4% | 1.2 (0.7 to 1.9) | 1.1 (0.7 to 1.7) | 0.64 |

Data shown are number (%), mean (standard deviation), or median (interquartile range).

*No significant differences (at *p* < 0.05) in percent missing data between trial arms.

Abbreviations: APCAPs, Andhra Pradesh Children and Parents study; BMI, body mass index; BP, blood pressure; HDL, is high-density lipoprotein; HOMA-IR, homeostatic model assessment-insulin resistance; IMT, is intima-media thickness; LDL, low-density lipoprotein; MET, metabolic equivalent task; SLI, Standard of Living Index

offspring until the age of 6 years (1.25 MJ energy and 8–10 g protein daily) was not associated with lower levels of cardiovascular disease risk factors (such as blood pressure, fasting blood lipids and insulin, arterial stiffness, and carotid intima media thickness) among the offspring when they became young adults (mean age 22 years).

**Table 3. Multivariable association between supplemental nutrition and cardiovascular risk factors of APCAPs participants in 2009–2012.**

| Outcome | N with data | Model 1 | | Model 2 | | Model 3 | |
|---|---|---|---|---|---|---|---|
| | | β (95% CI) | p-Value | β (95% CI) | p-Value | β (95% CI) | p-Value |
| Height (mm) | 1,783 | 1 (−5 to 8) | 0.70 | 2 (−5 to 8) | 0.63 | 1 (−5 to 7) | 0.70 |
| BMI (kg/m²) | 1,783 | −0.18 (−0.65 to 0.29) | 0.45 | −0.33 (−0.75 to 0.09) | 0.12 | −0.35 (−0.76 to 0.06) | 0.093 |
| Waist circumference (mm) | 1,779 | −5 (−15 to 6) | 0.40 | −8 (−17 to 1) | 0.100 | −9 (−18 to 1) | 0.065 |
| Systolic BP (mm Hg) | 1,782 | 1.0 (−0.3 to 2.2) | 0.13 | 0.5 (−0.6 to 1.6) | 0.34 | 0.5 (−0.6 to 1.6) | 0.36 |
| Diastolic BP (mm Hg) | 1,782 | 1.2 (−0.1 to 2.5) | 0.075 | 0.8 (−0.2 to 1.7) | 0.10 | 0.7 (−0.2 to 1.6) | 0.12 |
| Central systolic BP (mm Hg) | 1,395 | −0.1 (−1.2 to 1.0) | 0.88 | −0.2 (−1.2 to 0.9) | 0.73 | −0.2 (−1.2 to 0.8) | 0.72 |
| Pulse pressure (mm Hg) | 1,782 | −0.28 (−0.83 to 0.26) | 0.31 | −0.15 (−0.70 to 0.39) | 0.58 | −0.13 (−0.68 to 0.42) | 0.65 |
| Augmentation index (%) | 1,322 | −1.1 (−2.6 to 0.3) | 0.12 | −1.1 (−2.5 to 0.3) | 0.12 | −1.2 (−2.6 to 0.2) | 0.097 |
| Pulse wave velocity (m/s) | 1,542 | 0.05 (−0.03 to 0.13) | 0.25 | 0.03 (−0.04 to 0.10) | 0.39 | 0.02 (−0.04 to 0.09) | 0.48 |
| Carotid IMT (mm) | 1,194 | 0.01 (−0.02 to 0.03) | 0.62 | 0.01 (−0.01 to 0.03) | 0.32 | 0.01 (−0.01 to 0.03) | 0.36 |
| Total cholesterol (mmol/L) | 1,764 | 0.12 (−0.08 to 0.32) | 0.23 | 0.07 (−0.13 to 0.27) | 0.48 | 0.06 (−0.13 to 0.26) | 0.52 |
| LDL cholesterol (mmol/L) | 1,756 | 0.13 (−0.02 to 0.28) | 0.089 | 0.06 (−0.07 to 0.20) | 0.36 | 0.06 (−0.07 to 0.19) | 0.37 |
| HDL cholesterol (mmol/L) | 1,764 | −0.02 (−0.09 to 0.05) | 0.66 | 0.01 (−0.06 to 0.07) | 0.88 | 0.00 (−0.06 to 0.07) | 0.89 |
| Log triglycerides (mmol/L) | 1,763 | 0.03 (−0.03 to 0.08) | 0.34 | 0.02 (−0.04 to 0.07) | 0.53 | 0.02 (−0.04 to 0.07) | 0.58 |
| Fasting glucose (mmol/L) | 1,763 | −0.02 (−0.11 to 0.08) | 0.08 | −0.02 (−0.10 to 0.06) | 0.57 | −0.03 (−0.11 to 0.05) | 0.49 |
| Log insulin (mU/L) | 1,756 | 0.04 (−0.12 to 0.21) | 0.73 | −0.05 (−0.18 to 0.08) | 0.48 | −0.05 (−0.18 to 0.08) | 0.43 |

Model 1 is age and sex adjusted.

Model 2 is further adjusted for socioeconomic position and village urbanisation.

Model 3 is adjusted for variables in model 2 plus behavioural risk factors (tobacco and alcohol use; salt and fat consumption; and physical activity level and sedentariness).

Abbreviations: APCAPs, Andhra Pradesh Children and Parents study; BMI, body mass index; BP, blood pressure, HDL, high-density lipoprotein; HOMA-IR, homeostatic model assessment-insulin resistance; IMT, intima-media thickness; LDL, low-density lipoprotein; β (95% CI), beta coefficient and 95% confidence interval

## Strengths and limitations

The trial was conducted in an area with chronic undernutrition and homogeneous socioeconomic conditions, and women did not consume tobacco, limiting the potential for confounding by maternal and social factors. The subsequent unplanned urbanisation of the study villages, typical of peri-urban sprawls across many LMICs, allowed investigations into the effects of subsequent changes in environment, and make findings generalisable to transitioning areas of other LMICs. The loss to follow-up was relatively low in comparison to similar studies with long-term follow-up, limiting the potential for selection bias [15]. While out-migration for education and employment in young adulthood tends to be high, we may have benefitted from the proximity of the study villages to the urban centre, allowing many people to stay at home and commute to their place of work/study on a daily basis (>95% still reside in the village of their birth).

There are several limitations to this study that must be acknowledged. First, the villages in the trial were not randomised, raising the possibility of bias from other differences between trial areas. Although the control district was awaiting implementation of the intervention at the time of the trial (and did receive it a few years later in 1992–1993), the intervention district may have may have received the intervention earlier because of some other reason (e.g., political), which could be associated with other differences between the trial areas that could influence health outcomes. Second, we lacked data on compliance with the intervention. Anecdotal reports from investigators involved in the trial suggest that women in the trial were severely undernourished and collection of supplements was high, but the intake was not directly

**Table 4. Multivariable association between supplemental nutrition and cardiovascular risk factors in 2009–2012 in APCAPs participants who also presented in 2003–2005.**

| Outcome | N with data | Model 1 | | Model 2 | | Model 3 | |
|---|---|---|---|---|---|---|---|
| | | β (95% CI) | p-Value | β (95% CI) | p-Value | β (95% CI) | p-Value |
| Height (mm) | 863 | 6 (−2 to 14) | 0.12 | 7 (−1 to 15) | 0.095 | 7 (−1 to 15) | 0.089 |
| BMI (kg/m$^2$) | 863 | −0.43 (−0.90 to 0.05) | 0.080 | −0.50 (−0.94 to −0.06) | 0.027 | −0.51 (−0.98 to −0.05) | 0.031 |
| Waist circumference (mm) | 862 | −9 (−21 to 3) | 0.13 | −11 (−21 to −0) | 0.044 | −11 (−22 to −0) | 0.043 |
| Systolic BP (mm Hg) | 862 | 0.4 (−1.1 to 1.8) | 0.63 | 0.1 (−1.3 to 1.4) | 0.94 | 0.0 (−1.3 to 1.4) | 0.95 |
| Diastolic BP (mm Hg) | 862 | 0.8 (−0.4 to 2.0) | 0.20 | 0.4 (−0.9 to 1.6) | 0.56 | 0.3 (−1.0 to 1.5) | 0.69 |
| Central systolic BP (mm Hg) | 691 | −0.8 (−2.1 to 0.6) | 0.26 | −0.7 (−2.1 to 0.7) | 0.30 | −0.7 (−2.1 to 0.7) | 0.32 |
| Pulse pressure (mm Hg) | 862 | −0.4 (−1.1 to 0.4) | 0.32 | −0.3 (−1.0 to 0.5) | 0.49 | −0.2 (−0.9 to 0.6) | 0.66 |
| Augmentation index (%) | 663 | −2.2 (−3.7 to −0.7) | 0.005 | −2.3 (−3.8 to −0.8) | 0.002 | −2.3 (−3.7 to −0.8) | 0.002 |
| Pulse wave velocity (m/s) | 767 | 0.03 (−0.08 to 0.14) | 0.61 | 0.02 (−0.08 to 0.12) | 0.74 | 0.01 (−0.09 to 0.10) | 0.90 |
| Carotid IMT (mm) | 585 | −0.00 (−0.03 to 0.02) | 0.70 | −0.00 (−0.02 to 0.02) | 0.99 | 0.00 (−0.02 to 0.02) | 0.96 |
| Total cholesterol (mmol/L) | 855 | 0.08 (−0.14 to 0.30) | 0.49 | 0.02 (−0.20 to 0.23) | 0.88 | 0.00 (−0.21 to 0.22) | 0.96 |
| LDL cholesterol (mmol/L) | 852 | 0.07 (−0.11 to 0.25) | 0.42 | −0.02 (−0.17 to 0.13) | 0.80 | −0.02 (−0.17 to 0.12) | 0.75 |
| HDL cholesterol (mmol/L) | 855 | −0.00 (−0.08 to 0.08) | 0.97 | 0.02 (−0.06 to 0.10) | 0.63 | 0.02 (−0.06 to 0.10) | 0.63 |
| Log triglycerides (mmol/L) | 854 | 0.03 (−0.04 to 0.10) | 0.43 | 0.02 (−0.05 to 0.09) | 0.54 | 0.02 (−0.05 to 0.09) | 0.64 |
| Fasting glucose (mmol/L) | 855 | −0.03 (−0.13 to 0.07) | 0.59 | −0.01 (−0.11 to 0.09) | 0.81 | −0.02 (−0.12 to 0.08) | 0.68 |
| Log insulin (mU/L) | 854 | −0.05 (−0.21 to 0.11) | 0.54 | −0.12 (−0.26 to 0.01) | 0.073 | −0.14 (−0.27 to −0.00) | 0.045 |
| Log HOMA-IR | 854 | −0.06 (−0.22 to 0.10) | 0.46 | −0.13 (−0.26 to 0.00) | 0.056 | −0.14 (−0.27 to −0.01) | 0.030 |

Model 1 is age and sex adjusted.

Model 2 is further adjusted for socioeconomic position and village urbanisation.

Model 3 is adjusted for variables in model 2 plus behavioural risk factors (tobacco and alcohol use; salt and fat consumption; and physical activity level and sedentariness).

Abbreviations: APCAPs, Andhra Pradesh Children and Parents study; BMI, body mass index; BP, blood pressure, IMT, intima-media thickness; LDL, low-density lipoprotein; HDL is High-density Lipoprotein; HOMA-IR, Homeostatic Model Assessment-Insulin Resistance; β (95% CI), beta coefficient and 95% confidence interval

observed. External reports of the ICDS programme suggest that families often share the supplement with other children at home, which could have attenuated the effect of intervention. The lack of data on compliance raises the possibility that the intervention dose may not have been adequate to test the hypothesis robustly. There are no suitable data to guide adequacy of dose of nutritional supplementation, which would be conditional on usual diet from other sources. The ICDS scheme provides approximately 25% of daily energy and protein requirement for the women, and approximately 25% of energy and approximately 50% of protein requirement for the children. The 61 g (95% CI 18–104) difference in birth weight between the trial arms (intervention 2,655 g, control 2,594 g) compares favourably to the mean birth weight difference (41 g; 95% CI 4.7–77.3) reported in a systematic review of trials in which protein-calorie supplementation in pregnancy was directly observed, suggesting that most of the supplement was consumed by the intended beneficiary [12,13,21].

Third, about a third of the cohort could not be followed up. Simple comparison of data collected at the time of first follow-up showed no material difference between those who took part or did not take part in the second or third follow-ups (Table 1); nevertheless, potential bias in results due to systematic differences in characteristics of those lost to follow-up cannot be ruled out. Fourth, participants and observers who collected field data were not 'blind' to the intervention allocation, which could introduce bias; however, data collection on several outcomes was either automated or processed by those who were blind to the allocation of intervention (e.g., biochemical assays), reducing the likelihood of such bias. Finally, given the age of

the participants and the rural study setting, the levels of cardiovascular disease risk factors were relatively low, reducing our ability to detect differences. As rates of cardiovascular disease tend to increase more rapidly with age in India than in many other countries, further follow-up of participants in mid-adulthood will generate more robust data on incident cardiovascular disease [22].

## Comparison with previous research

There are few trials of protein-calorie nutritional supplementation offered to pregnant women and young children with follow-up for cardiovascular risk in adulthood. A cluster randomised trial in The Gambia (28 villages; $N$ = 1,317) provided protein-calorie supplements (4.25 MJ of energy and 22 g protein daily) to pregnant (intervention, from 20-week gestation until delivery) or lactating (control, from 20-week gestation until delivery) women, and found no difference in cardiovascular risk factors (blood pressure, body composition, blood cholesterol, and fasting glucose) among offspring when they were aged 11–17 years [10]. Another cluster randomised trial from Guatemala (4 villages; $N$ = 429) offered 3.8 MJ of energy and 64 g protein (intervention) or 1.4 MJ of energy (control) supplements twice daily to pregnant and lactating women and children until the age of 7 years and found no difference in cardiovascular risk factors among offspring when they were aged 24 years; however, in subgroup analyses, intervention between the ages of 3 and 6 years was associated with lower fasting glucose and systolic blood pressure, while intervention between birth and 3 years was associated with lower triglyceride and higher high-density lipoprotein cholesterol levels [11]. A trial from rural Bangladesh offered protein-calorie supplementation to pregnant women in early or late pregnancy; at a follow-up conducted at age 4.5 years, early supplementation was associated with lower diastolic blood pressure, but findings in adulthood have not been reported [23].

At the time of the first follow-up of the APCAPs cohort, nutritional supplementation was associated with beneficial effects on several cardiovascular risk factors (i.e., height, arterial stiffness, and insulin resistance) (Fig 3) [12]. There are several potential explanations for the difference in findings between the first follow-up (mean age 16 years) and the second or third follow-up (mean age 22 years) reported here. Assuming a true null as the most likely explanation, the positive findings in the first follow-up could have been due to chance (particularly since multiple statistical tests were conducted) or selection bias (participants from the intervention villages with favourable outcomes selectively presenting for examination at the first follow-up). There was weak evidence in favour of the latter hypothesis in the present analyses when the sample was restricted to offspring who were also part of the first follow-up; borderline associations were seen for relevant outcomes, although the overlapping confidence intervals were consistent with no difference between follow-ups (Tables 3 and 4). Alternatively, a true effect of the intervention in adolescence may been attenuated over the long-term follow-up due to nutrition or economic transition [6,7], which would support the view that the risk of cardiovascular disease accumulates across the life course [17]. Between 1991 and 2011, the population of study villages increased by 31% (census data), and NTLI and built-up land use (both from remote sensing data) increased by 156% and 196%, respectively, confirming urbanisation of the study villages, which was also found to be associated strongly with cardiovascular risk factors in APCAPs participants [24]. There was some support for this hypothesis from models stratified by urbanisation, which showed weak evidence of the effect of intervention in the least urbanised villages (S3 Table). However, future follow-ups of the participants will be able to provide much greater clarity as the cardiovascular phenotypes mature with age.

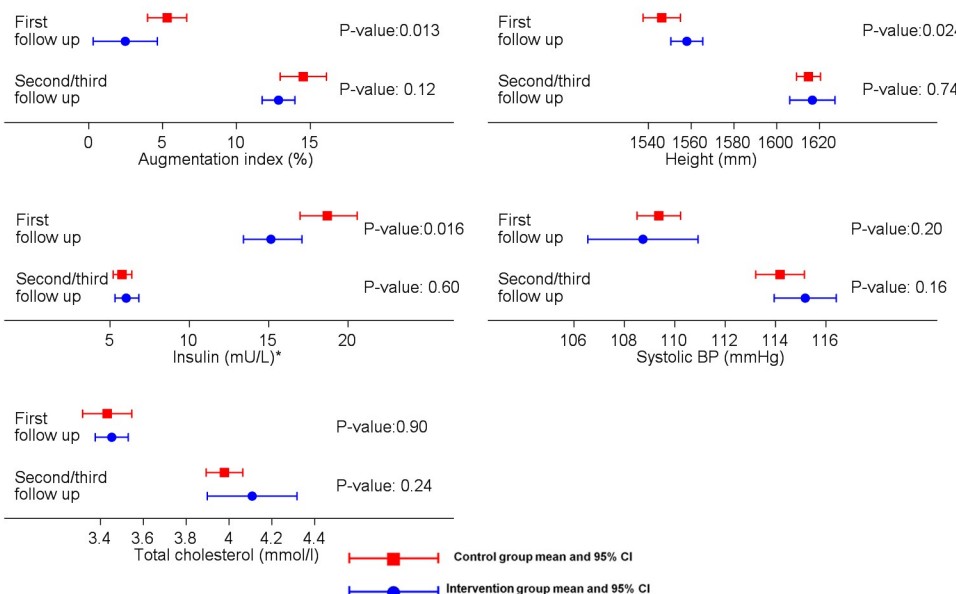

**Fig 3. Comparison of findings for selected outcomes in APCAPs participants presenting at first follow-up (FU1, mean age 16 years) and second or third follow-up (FU2/3, mean age 22 years).** Data shown are means and 95% confidence intervals (blue bars, control; red bars, intervention). *Geometric mean was used for this variable. APCAPs, Andhra Pradesh Children and Parents study; BP, blood pressure.

## Implications

Cardiovascular disease is the leading cause of death and disability in LMICs [1]. The Developmental Origins of Health and Disease hypothesis is of great importance to the unfolding epidemic of cardiovascular disease in many LMICs undergoing nutrition transition, notably countries of South Asia (home to a quarter of the world's population), where up to a quarter of all births are low birth weight [25,26]. These data could not confirm a clear benefit of balanced protein-calorie food supplementation given to undernourished pregnant women and children on cardiovascular disease prevention. Conversely, there was no increase in cardiovascular risk associated with nutritional supplementation, which is important for similar food supplementation programmes in many LMICs that are now concerned about the emerging epidemics of obesity and cardiovascular disease [27]. Given the study limitations and young age of the participants, adverse cardiovascular effects of undernutrition in pregnancy and childhood cannot be ruled out; however, data from trial evidence to date do not support the 'real-world' effectiveness of balanced protein-calorie interventions in pregnancy and/or childhood. There are other potential benefits of improved child nutrition, including respiratory function, cognition, schooling and labour market participation, and reproductive health, which have also been noted in APCAPs [20]. However, until further evidence is available, policy makers should attach limited value to cardiovascular health benefits of maternal and child protein-calorie food supplementation programmes. Meanwhile, further research involving long-term follow-up of participants of intervention studies in which the compliance is monitored is needed to provide more conclusive evidence.

## Conclusions

In an area with prevalent undernutrition, protein-calorie food supplementation offered to pregnant women and their offspring in childhood did not lower the offspring's risk of

cardiovascular disease in young adulthood. While adverse cardiovascular effects of undernutrition in pregnancy and childhood cannot be ruled out, data from intervention studies to date do not support the real-world effectiveness of balanced protein-calorie interventions in pregnancy and/or childhood.

## Supporting information

**S1 CONSORT Checklist.**
(DOCX)

**S1 Text. Study protocol for APCAPs second and third follow-ups.** APCAPs, Andhra Pradesh Children and Parents study.
(PDF)

**S1 Table. Age-and sex-adjusted effect of supplemental nutrition by age at outcome measurement.**
(DOCX)

**S2 Table. Age-adjusted effect of supplemental nutrition on outcome by sex of the participant.**
(DOCX)

**S3 Table. Effect of supplemental nutrition by village urbanisation score.**
(DOCX)

**S4 Table. Effect of supplemental nutrition by BMI of participant.** BMI, body mass index.
(DOCX)

## Acknowledgments

We wish to acknowledge our dedicated field teams and the study participants who made this study possible.

## Author Contributions

**Conceptualization:** Sanjay Kinra, Poornima Prabhakaran, Vipin Gupta, George Davey Smith, K. V. Radha Krishna, Shah Ebrahim, Hannah Kuper, Yoav Ben-Shlomo.

**Data curation:** Santhi Bhogadi, Poppy Alice Carson Mallinson.

**Formal analysis:** John Gregson.

**Funding acquisition:** Sanjay Kinra, Poornima Prabhakaran, George Davey Smith, K. V. Radha Krishna, Shah Ebrahim, Hannah Kuper, Yoav Ben-Shlomo.

**Investigation:** Sanjay Kinra, Poornima Prabhakaran, Vipin Gupta, Gagandeep Kaur Walia, Dorairaj Prabhakaran, George Davey Smith, K. V. Radha Krishna, Shah Ebrahim, Hannah Kuper, Yoav Ben-Shlomo.

**Methodology:** Sanjay Kinra, Poornima Prabhakaran, Vipin Gupta, Gagandeep Kaur Walia, Santhi Bhogadi, Ruby Gupta, Bharati Kulkarni, K. V. Radha Krishna, Shah Ebrahim, Hannah Kuper, Yoav Ben-Shlomo.

**Project administration:** Sanjay Kinra, Poornima Prabhakaran, Vipin Gupta, Gagandeep Kaur Walia, Santhi Bhogadi, Ruby Gupta, Aastha Aggarwal, Bharati Kulkarni, K. V. Radha Krishna, Shah Ebrahim, Hannah Kuper.

**Resources:** Bharati Kulkarni, K. V. Radha Krishna, Shah Ebrahim.

**Supervision:** Sanjay Kinra, George Davey Smith, Shah Ebrahim, Hannah Kuper, Yoav Ben-Shlomo.

**Visualization:** Sanjay Kinra.

**Writing – original draft:** Sanjay Kinra.

**Writing – review & editing:** Sanjay Kinra, John Gregson, Poornima Prabhakaran, Vipin Gupta, Gagandeep Kaur Walia, Santhi Bhogadi, Ruby Gupta, Aastha Aggarwal, Poppy Alice Carson Mallinson, Bharati Kulkarni, Dorairaj Prabhakaran, George Davey Smith, K. V. Radha Krishna, Shah Ebrahim, Hannah Kuper, Yoav Ben-Shlomo.

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
