## [Editor Report · Decision Letter 0]

20 Jan 2020

Dear Dr Kinra, 

Thank you for submitting your manuscript entitled "Impact of nutritional supplementation in pregnancy on offspring’s risk of cardiovascular disease: long-term follow-up of a cluster trial" for consideration by PLOS Medicine.

Your manuscript has now been evaluated by the PLOS Medicine editorial staff [as well as by an academic editor with relevant expertise] and I am writing to let you know that we would like to send your submission out for external peer review.

Kind regards,

Adya Misra, PhD,

Senior Editor

PLOS Medicine

---

## [Decision Letter · Decision Letter 1]

17 Mar 2020

Dear Dr. Kinra,

Thank you very much for submitting your manuscript "Impact of nutritional supplementation in pregnancy on offspring’s risk of cardiovascular disease: long-term follow-up of a cluster trial" (PMEDICINE-D-20-00167R1) for consideration at PLOS Medicine. 

[LINK]

In light of these reviews, I am afraid that we will not be able to accept the manuscript for publication in the journal in its current form, but we would like to consider a revised version that addresses the reviewers' and editors' comments. Obviously we cannot make any decision about publication until we have seen the revised manuscript and your response, and we plan to seek re-review by one or more of the reviewers. 

We expect to receive your revised manuscript by Apr 07 2020 11:59PM. Please email us (plosmedicine@plos.org) if you have any questions or concerns.

We look forward to receiving your revised manuscript. 

Sincerely,

Adya Misra, PhD

Senior Editor 

PLOS Medicine

plosmedicine.org

Comments from the Academic Editor:

The authors examined the intervention effects of food supplements in a unique trial conducted in three decades ago on profiles of cardiovascular risk factors. 

Main comments: 

1. The reviewer initially somehow read as if this study were a community-based cluster randomized controlled trial. It appeared not, where the program did not randomly determine intervention villages and control villages (confirmed in Ref #12, BMJ article). The non-randomized design should be apparent in Abstract at least. Also, the authors should state the selection criteria for the intervention and no intervention explicitly in the first paragraph of Methods. The authors are encouraged to state clearly, in the abstract and the first paragraph of the Methods, that this study was a community-based non-randomized controlled intervention trial.

2. The authors found null associations in the primary analysis and described why in Discussion. The authors described the reason that the null finding might be due to the confounding effects of puberty or related factors. This explanation does not make sense. The confounding factor must have occurred before the exposure status was determined. The primary exposure was the intervention to mothers of the participants and to the participants themselves during their childhoods. Puberty factors could not be confounders.

The possible explanations should include the true null, flagging up the possibility of by-chance findings in the previous analysis (BMJ). This current study, showing Table 3 and 4, indicated the selection bias in the previous analysis. The participants selectively showed up the first follow-up and presented favourable outcomes, thereby leading to potentially false-positive results. Also, the previous analysis in BMJ demonstrated a lot of statistical tests and may have substantially raised the likelihood of false-positive findings: the authors should take a fresh look at the results with the null. The authors should suspect such possibilities. 

Then, the authors may want to highlight the overall cohort design, given the non-random nature of the exposure and also non-random nature of the responses to the follow-up assessments. It would be difficult to solve, but that should be the point to highlight.

3. The other possible explanation of the findings was that any positive or adverse effect of the intervention might get diluted over the long-term follow-up given the nutrition or economic transition over time. The authors are encouraged to extract such information from regional data.

4. The authors should explore the literature and discuss whether the dose levels were sufficient to test the hypothesis. 2.5 MJ of energy intake and 20 g of protein on a single day or over a few months (frequency and duration were not available in this paper actually) may not have been sufficient. The authors should look at the requirement of energy intake and protein intake for pregnant women and the toddlers and compare between those required levels and the provided levels via food supplements.

It would be acceptable that the authors discuss their speculation. However, before the speculation (e.g. puberty effect) without physiologic explanation, the authors seem to have a lot to discuss.

5. The authors should confirm or disconfirm that the intervention villages were selected because of their socially deprived condition. If so, the authors were testing whether the food supplements let people in deprived villages catch up with those in more affluent villages. This interpretation would be optimistic but fair. The authors may want to touch on this while no randomization was certainly the limitation and to be on emphasis much more. 

At the same time, the authors describe the difference between pregnant women in intervention villages and control villages.

Minor comments: 

Abstract:

"cluster trial" should be "cluster non-randomized trial". 

Introduction:

"maybe" should be "may be" in the end of the first paragraph.

The sentence "We previously reported..." should be taken out.

The abstract should include limitations of no randomization or residual confounding and no assessment of compliance with the intervention at least. 

Levels of dose, frequency, and duration of the intervention should be available in Abstract.

The conclusion requires revision. The authors should not give a conclusion about what this cohort will have to do: readers may support the promise of this study, but that cannot be the conclusion of this study. The last sentence should be taken out. 

Methods:

"South India" sounds like a country. "India" seems fine.

On the third line of the first paragraph, "aims" may sound too odd because the trial started long ago and "ICDS" is not clear for readers (and the reviewer) whether it has been lasting for a long time. The authors may want to supplement the information that ICDS is a long-term scheme lasting over decades in India to let the tense make sense.

The authors should clarify the dose frequency and duration of the food supplement, not only the dose levels. Readers could interpret that the investigators had provided just a single package of food.

In the second paragraph of Methods, "Where individual participant data was" should be "Where individual participant data were". 

The authors stated assay quality, but IMT and some of the other measures are not assays. The authors need to rephrase or add sufficient information on a measurement quality for different physiologic measures.

Here are some comments about statistical analyses.

"We fitted a random-effect at both village and household levels" is not clear. The authors could assign a random-effect to different parameters in a regression model. Those could be an intercept, a beta coefficient for the binary intervention variable, or both. The current description is not clear. It is highly important to test or describe the variation of effects across villages. 

In the statistical section, the authors should explicitly state the need for the statistical adjustment because the trial was susceptible to the bias without randomization.

It would be meaningful to adjust for mediating or intermediary factors. However, the factors the authors describe just after the thought should not be age and sex. They cannot be mediating factors. The authors may want to revise the order of the description would allow an odd interpretation. Alternatively, the authors should state what they think are mediating factors.

Results:

The authors should not abbreviate 2012 into 12.

Table 1 must include information stratified by trial arms: intervention arm and control arm. 

Table 2 should have a title of the year or year range of these characteristics and then provide a footnote to explain the timepoint was follow-up measures.

Table 2 has two columns of the sample sizes for different rows. The authors may better drop this. Then, after placing the maximal numbers of participants of the two arms (949 and 877), create the following two columns.

Average % of missing information. Use a superscript to indicate whether the percentages were statistically significantly different between arms.

P values for differences between arms.

In Table 2, the authors may better indicate that "Resident village urbanization" and "Standard of living index" were both scores based on some index. Otherwise, "third" would not make sense. 

Table and Figure. For each p-value higher than 0.1, present all the values up to the second decimal place.

Table S3. CIMT needs an explanation for what it stands for.

"Table S1-4" is confusing. The authors need to revise this.

The authors use "wave 1" or "wave 2 and 3", but it is confusing. Descriptions in text and tables are not consistent (sometimes in "wave" and other times in years), and it is difficult to read. Then, additionally, in Figure 2, the authors state FU1 and FU2/3. All are confusing. The authors should state "baseline", "follow-up 1", and "follow-up 2&3" throughout in text, tables, and figures. "Wave" is not a proper scientific term. 

Discussion:

The interpretation of Table 4 in comparison to that of Table 3 should include the selection bias during follow-up. The results in the subset are sensitive to the sampling, which is a study-specific factor. The argument about puberty is little convincing, without any biologically plausible, persuasive arguments. The authors are highly encouraged to take out the argument.

The argument about confounding should be related to the selection of village in regards to the definition of confounding (determinants of exposure). The authors should dig into the selection criteria in details. It is just not convincing (or non-scientific) to speaking about the factors after the intervention as a source of confounding.

The authors should discuss the lack of blinding in this trial for the participants and the assessors. The reviewer understands it would be difficult to achieve, but the limitation should be available in the text.

The authors should describe the policy implications that many interventions may be carried out without any apparent effects on cardiovascular phenotypes.

The lack of assessment of compliance with the intervention seemed to be crucial. The authors' view was too optimistic. The difference by 61 g in offspring's birth weight may have been too small given the absolute levels of birth weights; and also could be by chance, given the history of multiple tests in the prior publications. 

Abstract- please provide participant demographics

Abstract- please provide clear number of years of follow-up reported previously and those reported in the current submission

Please revise for clarity and grammar “The levels of cardiovascular risk factors were generally low” and also avoid the use of vague language like generally

Please provide p-values along with 95% CI throughout and within abstract

Abstract methods and findings-last sentence should be a limitation of your methodology

Abstract conclusions should begin with “our results show” or similar 

Please place full stop after the reference square brackets

Introduction second paragraph- please provide a sentence or two about the previous study before you mention trial and control villages as it is currently not obvious that the previous study was in the same population as in the current submission

Was this trial registered? Please provide details, either way within the methods section

Please provide the study protocol as supplementary information.

Please revise the reporting in accordance with CONSORT guidelines and provide a completed CONSORT checklist as supplementary information

Telangana is a large state, please consider providing a map of the villages in both arms to highlight the spread as well as to indicate which areas have undergone urbanisation 

Please clearly provide names and citations of all surveys, questionnaires used in this study. “Tobacco and alcohol were assessed by standard questions” is not sufficient.

Please provide participant demographics in the methods section

Please provide details of ethics approval and informed written consent in the methods section

Discussion- please summarise findings in one paragraph, followed by strengths, limitations and comparison to other studies in the literature

Comments from the reviewers:

Reviewer #1: This is a statistical review of manuscript PMEDICINE-D-20-00167_R1. The manuscript reports the results of a long-term follow-up of a cluster trial in nutritional supplementation during pregnancy on offspring's risk of cardiovascular disease. 

The statistical methods are appropriate. The reporting is clear, especially thanks to the use of Tables. I only have two minor comments related to Figure 2. Firstly, the text mentions that height was different in adolescence but height isn't reported on Figure 2. Secondly, a legend is missing to distinguish what the blue and red bars represent. 

Reviewer #2: This is an interesting manuscript, describing the follow up of young adults born during a village-based nutrition intervention programme in pregnancy and early childhood in Hyderabad, India. It is an important contribution to the DOHaD field. 

Specific comments below largely relate to missing detail about the study design, cohort details etc. that prevent the reader from having a full understanding of the cohort. 

Abstract 

Background: Developmental Origins of Adult Disease Hypothesis is now more commonly called the Developmental Origins of Health and Disease hypothesis. Also in Background, paragraph 1, and Discussion. 

Insert word, 'control of THE cardiovascular disease epidemic ….'. 

Methods and findings, insert word 'further surveys of THE offspring ….'

Define in abstract, is the 1826 70% of the original number of mother-infant dyads randomized? In the method, it states that 2,964 infants were born into the study (or had a birthweight measured) - 1826/2964 = 61%. This isn't clear. 

Add some detail to define the low cardiovascular disease risk in the adults (e.g. mean BMI, level of elevated blood pressure). 

Background

Missing word "social disadvantage in THE early years of life ….."

Methods

Paragraph 2. Further confusion around participant numbers. In paragraph 1, it says that in 2003-5, N=2601 families were contacted (unless I'm misreading this). But in paragraph 2, it states that N=1165 were followed up in 2003-5. Please make this clear. 

What was the difference between Wave 2 and Wave 3? Why were there two separate waves, one year apart? Was there any order in how participants were retraced during these waves?

I note that ten years has passed since wave 2 was complete. Should this be mentioned, e.g. in relation to any further secular trends with respect to CVD risk in India?

Measurements. The reader is referred to two other papers for details on the methodologies used (refs 12 and 14). Ref 12 is the previous follow up study on adolescents, conducted several years earlier. Were all the same interviews used? Is this appropriate, given the differing age/context of the adolescent/young adult groups? Please be specific about which questionnaires/measurements were used as described previously, and which were implemented specifically in this study of adults. 

Weight was measure by digital weighing scales (not machine). 

For anthropometry, I assume measurements were made in duplicate as you indicated that "The mean of two anthropometric measurements …..". But please make this clear. Also, if values were different by <5cm etc. were both values rejected. Or was this at the point of measurement? Please be clear. 

Dietary intake data was collected using a semi-quantitative FFQ: "A semi-quantitative food frequency questionnaire coupled with customized nutrient databases was used to estimate average daily salt and fat consumption and energy intake; its validation against multiple weighed 24-hour recalls in this setting has been published.[14]". Only salt and fat intakes have been selected for presentation in the current analysis. I think a justification on this point would be helpful. Also, how are salt intakes estimated from a FFQ? Dietary assessment of sodium intake is unreliable at best, even from a full weighted record. A comment on why these two components of dietary intake have been used in the analysis and their reliability, is warranted. Why not intakes of fruits and vegetables? Or sugar sweetened beverages? Would be interesting to look at other components of diet, given the transitional nature of the population. 

In addition, Reference 14 is the COHORT profile paper, and does not describe the validation of this tool. Similarly for the statement on physical activity assessment. Please provide the correct, original reference. 

Insert word, "AN enzymatic calorimetric method ……"

Page 13, the method selected for assessing 'village urbanisation' (night-time light intensity) has been shown to be valid in a single paper from China. Are there any other internal modes of validation that could be used from within the existing dataset to demonstrate the robustness of this tool within the Indian context?

Ethics. On Page 12 you describe PPI. Where and how was ethical approval obtained for the follow-up survey described here?

Results. 

Table 1. Add sex distribution to Table 1. Is the 62% male follow up reported in Table 2 a similar bias (towards males) as in the original birth cohort and in Wave 1?

The intent of this sentence is not clear "About 20% of the intervention arm participants lived in the least urbanised third of villages, as opposed to 44% from the control arm." Or rather, I understand the data as presented in Table 2 - but it's not clear if this reflects a bias in rate of urbanisation between the control and intervention villages. I think it would be useful to include this data also in the results - e.g. not participant based, but cluster based. What is the distribution of original village clusters according to the new Resident village urbanisation category? Or are we to assume that participants have not moved - so where they live now, is the same village they lived in when born into the study?

Indeed, in the Discussion, you state that "then the subsequent attenuation of this difference may have resulted from environmental pressure of urbanisation (which has been disproportionately more in the intervention villages)" - but this is not clear from the results. Please modify. 

Figure 2. I think it would be more helpful to the reader to replace the axes labels of 'FU1' and 'FU2' with the age of the participants at follow up. 

Discussion

An important omission is the MINIMat trial in Bangladesh, which was a trial of early or usual food supplementation (+ additional micronutrients). Cardiovascular disease risk factors have been measured in the offspring. 

https://www.ncbi.nlm.nih.gov/pubmed/23514767

https://www.ncbi.nlm.nih.gov/pubmed/23241449

Also, an important difference to note between the current Indian trial and those reported from The Gambia and Guatemala (< Bangladesh should be added) is the timing of intervention; India and Guatemala were interventions given in pregnancy and early childhood; The Gambia and Bangladesh supplements were given in pregnancy only. You should add some consideration of the results obtained from the current study, in relation to these other studies. The Discussion on this point is rather cursory. 

I do not follow the statement that "The linkage to additional data collected during the trial (e.g. nutritional supplement collection and maternal and offspring anthropometrics) was not reliable enough to be incorporated in these analyses". It is not clear why linkage to these data is not reliable, yet linkage to other participant details (village of birth, dates of birth) is considered reliable. A greater description on the fidelity of the participant information needs to be included in the methods section of the current paper. Is the data presented on birthweights (reported in both the introduction and discussion, published elsewhere? If so, please include a reference. If not, some further information should be included in the Results of the current paper (birthweight by village, sex). 

I also don't understand this statement: "We previously analysed the birthweight data from the trial (birthweights were recorded separately for each village, despite unreliable linkage to study participants) and found a difference of 61g (95%CI 18 to 104g) between trial arms (intervention 2655g, control 2594g), which is comparable to birthweight difference (mean difference 41g, 95% CI: 4.7 to 77.3) reported in a systematic review of trials in which protein-calorie supplementation in pregnancy was directly observed, suggesting that this is a minor concern.[12,13,22]". What is a minor concern? This needs restructuring to make it clear what is meant. 

Were there any differences in current CVD risk factors according to level of urbanisation of the villages? I believe this could be of value in terms of (i) demonstrating the robustness of the urbanisation tool and (ii) explaining the lack of any associations observed. 

Reviewer #3: PMEDICINE-D-20-00167R1

Impact of nutritional supplementation in pregnancy on offspring's risk of cardiovascular disease: long-term follow-up of a cluster trial

Thank you for the opportunity to review this manuscript. This is an interesting and compelling revised paper from a strong research team. The authors assessed whether the provision of supplemental nutrition to undernourished pregnant women and their offspring lowered the offspring's risk of cardiovascular disease in young adulthood. The findings of the analyses were null (and the author conducted numerous analyses) but this does not detract from the paper's importance. The authors note that rates of cardiovascular disease tend to increase more rapidly with age in India (more so than in many other countries) and so further follow-up of offspring into middle adulthood may yield different results.

This paper is well written and will be of interest to maternal and child health practitioners and researchers interested in early life origins of adult disease. 

Minor comments:

1. The Abstract does not include the primary research question or aim of the study. As written, the research question is not clearly stated but may be gleaned from the middle of the Methods and findings section. The research question is also not clearly stated in the Background section of the manuscript. It would be helpful to include a statement describing the aims of the study just prior to presenting the hypothesis.

2. The funding disclosures presented at the beginning and end (within the body) of the manuscript are somewhat inconsistent. 

3. There are inconsistent decimal places for estimates and Cis in Tables 3 and 4 and p values presented in supplementary tables.

[LINK]

---

## [Decision Letter · Decision Letter 2]

22 May 2020

Dear Dr. Kinra,

Thank you very much for re-submitting your manuscript "Effect of supplemental nutrition in pregnancy on offspring’s risk of cardiovascular disease in young adulthood: long-term follow-up of a cluster trial" (PMEDICINE-D-20-00167R2) for review by PLOS Medicine.

I have discussed the paper with my colleagues and the academic editor and it was also seen again by xxx reviewers. I am pleased to say that provided the remaining editorial and production issues are dealt with we are planning to accept the paper for publication in the journal.

[LINK]

We look forward to receiving the revised manuscript by May 29 2020 11:59PM. 

Sincerely,

Adya Misra, PhD

Senior Editor 

PLOS Medicine

plosmedicine.org

Requests from Editors:

Title- Please add India to the title. Could you please also just say "follow-up of a cluster trial". 

The original manuscript document instead of the revised document has been included in the manuscript PDF, please remove the original manuscript copy within editorial manager 

Author summary- please add bullet points

Line 66,67 needs revision, since you are presenting results from a non-randomised study? 

Lines 75,76- could you revise for clarity? The use of “who” twice is a bit unclear

Author summary lines 78-79 should be tempered, as there are several confounders. Perhaps you can add “our results suggest…” or similar?

Line 80- “cannot” is rather emphatic, please tone down to “should not”. This part of the summary should be revised to indicate that an intervention to the limited time segments (just during a pregnancy period and just after giving birth) was insufficient.

We do not require a patient and public involvement section. However, if you wish to include any of these details within the methods section please do so

Please can you confirm that the included map is free from any copyright restrictions? 

Please state within the methods section if any changes were made to the study protocol and provide reasons for the same 

Could you please remove page numbers in the CONSORT checklist and replace with section and paragraph numbers. Please also rename the file to state that it is the CONSORT checklist.

Implications

Implications should include the need for monitoring, in general, compliance of program participants with an intervention program.

It is possible that a lack of monitoring would make an effective intervention meaningless. This study is in line with it.

Table 2-The authors should create one more column of p-values from the test comparing intervention groups and control groups for each variable.

Or I would suggest revise this column to include p-values from the comparison between two groups. It may be ok to put a table footnote that there was no significant difference in degrees of missing information between the two groups.

Village urbanisation score This variable clearly showed the difference between the intervention group and control group. The authors should highlight it.

Table 3- I would suggest to create two columns for Model 1, Model 2, and Model 3 each, putting effect measures in one column and p-values in the other column in each. “p=0.xx”look too messy.

Fig3 Bar charts are not good choice. A bar indicates a range from 0 to some positive value. But none of the variables in Fig 3 can be zero. The authors should use diamonds or circules as the best point estimates and 95% confidence intervals.

Comments from Reviewers:

Reviewer #1: The authors have satisfactorily addressed my comments. 

Reviewer #2: The authors have adequately addressed all the comments made by both myself and the other reviewers and I believe this paper is now suitable for publication.

[LINK]

---

## [Editor Report · Decision Letter 3]

16 Jun 2020

Dear Dr. Kinra, 

On behalf of my colleagues and the academic editor, Dr. Fumiaki Imamura, I am delighted to inform you that your manuscript entitled "Effect of supplemental nutrition in pregnancy on offspring’s risk of cardiovascular disease in young adulthood: long-term follow-up of a cluster trial from India" (PMEDICINE-D-20-00167R3) has been accepted for publication in PLOS Medicine. 

PRODUCTION PROCESS

PRESS

PROFILE INFORMATION

Thank you again for submitting the manuscript to PLOS Medicine. We look forward to publishing it. 

Best wishes, 

Adya Misra, PhD

Senior Editor 

PLOS Medicine

plosmedicine.org